# Urban Stormwater Resilience Assessment Method Based on Cloud Model and TOPSIS

**DOI:** 10.3390/ijerph19010038

**Published:** 2021-12-21

**Authors:** Han Qiao, Jingjing Pei

**Affiliations:** School of Engineering & Technology, China University of Geosciences, Beijing 100083, China; 2102200047@cugb.edu.cn

**Keywords:** resilience, indicator, evaluation system, cloud model, TOPSIS

## Abstract

To scientifically and quantitatively evaluate the degree of urban storm resilience and improve the level of urban stormwater resilience, based on the resilience theory, starting from the three attributes of resilience (resistance, recovery and adaptability), this paper establishes the framework of urban resilience evaluation indicator system under the background of stormwater disaster. Firstly, the weight of the indicator system is analyzed by the Delphi method and cloud model, and then the urban stormwater recilience evaluation model is constructed by using the cloud model and approximate ideal solution ranking method. Through the fuzzy description, the corresponding quantitative value is given to the qualitative indicator, so that the stormwater resilience of the city can be measured by accurate values. Finally, the feasibility of the model is verified by case analysis. The results show that the urban stormwater resilience evaluation theory and method based on cloud model and approximate ideal solution ranking method have important guiding significance to improve the level of urban stormwater resilience.

## 1. Introduction

In recent years, the impact of climate change on cities has become more and more serious. In particular, natural disasters such as heavy rain and flood brought by extreme weather have posed a huge threat to the public safety of cities around the world [1]. In such a context, the research related to resilience science opens up a new perspective for the study of urban security. Among them, the evaluation of urban stormwater resilience is a major research hotspot. Based on the basic theory of resilience, this research evaluates the ability of cities to deal with the flood damage from various aspects and dimensions, finds out the weaknesses in each structure and link of the city, and finally puts forward the corresponding improvement suggestions and measures [2].

With the deepening of the research on the discipline of resilience, scholars from all over the world began to refine the evaluation of urban stormwater resilience into specific indicators, and scientifically evaluate the degree of urban stormwater resilience with quantitative indicators, making the design of an urban stormwater resilience evaluation system more comprehensive and scientific. For example, in 2013, Orencio and other scholars constructed a resilience index system for coastal communities to cope with flood disasters and elaborated the corresponding seven aspects that affect the resilience of coastal communities [3]. Then, the AHP method (analytic hierarchy process) was used to calculate the resilience index of cities. In 2016, Kotzee and Reyers proposed a comprehensive index covering economic, social, ecological, and infrastructure indicators related to stormwater resilience, and mapped the spatial distribution of resilience from the perspective of system adaptability and vulnerability [4]. In the same year, Qasim and other scholars established a four-layer community flood disaster resilience index of social, economic, institutional and physical [5], and evaluated and analyzed the community resilience index in three districts of Khyber Pakhtunkhwa Province by using expert evaluation method. In 2017, Miguez and Verol designed a stormwater toughness index model tool with the help of the multi-criterion method and hydrodynamic model, which can quantitatively express the toughness index of different stormwater management schemes [6]. Yu Kongjian proposed three key strategies for stormwater resilience urban construction: absorption, deceleration, and adaptation, and verified their effectiveness through field experimental projects [7]. According to the characteristics of Weimar Ilmpark, Tian Lu analyzed the necessity and potential of stormwater management in this area and proposed corresponding improvement strategies for the construction of stormwater resilience cities in China based on urban resilience theory [8]. Zhou Yinan et al. constructed a set of storm flood resilience urban design systems, which includes the optimization of land use, urban organizational structure, establishment of multi-functional space and integration of an urban system, extending the boundary of traditional urban design and providing a new choice for performance eco-city design [9]. Ma Kun et al. constructed a stormwater management model based on the toughness theory, and then proposed a corresponding stormwater management construction model by calculating the flood-bearing capacity of the study area [10]. The research on urban storm flood resilience started late in China, and the assessment of urban storm flood resilience mainly focuses on qualitative research.

In this paper, the urban stormwater resilience assessment system is taken as the research object, and the influence degree of natural disasters and safety production technology level on the urban stormwater resilience is mainly studied. Therefore, the purpose of this paper is to improve the city’s ability to resist natural disasters by effectively evaluating the resilience of urban rainwater, so as to provide a theoretical basis for strategic thinking and countermeasures for the construction of resilient cities.

## 2. Related Theories and Methods

### 2.1. Resilience and Stormwater Resilience

Resilience means “rebound or retreat.” In the 1970s, Canadian ecologist Hollin first applied the concept of resilience in the field of systems ecology to express “the ability of an ecosystem to recover to its original structural stability after being impacted by the outside world” [11]. With the deepening of relevant research and interdisciplinary exchanges, the concept of resilience has gradually expanded to the fields of human beings, social ecology, urban economy, urban and rural planning. At the same time, scholars’ interpretation of the connotation of resilience is becoming more and more thorough and perfect. The academia generally believes that resilience has three basic elements, namely, the system’s defense ability to resist external shocks, the resilience ability to recover to its original structure after bearing external disturbances, and the better adaptability to withstand the risk impact of the next disaster.

The concept of stormwater resilience originates from system ecology and is a kind of resistance, resilience, adaptability and self-learning ability to cope with extreme natural disasters such as flood and waterlogging, with a certain degree of redundancy [12]. The concept and connotations of stormwater resilience are not uniform and fixed, and mostly change with the change in practice and application objects [13]. For example, Bell believes that the stormwater resilience of buildings refers to the ability of buildings to withstand the damage of rain and flood [14]. Tourbier believes that storm flood resilience in planning spatial dimension can be summarized from four aspects: society, structure, space and risk management, mainly including flood control engineering, green space management, institutional management, flood area planning, storm runoff control and financial insurance assistance [15]. Batica et al. believed that regional or community storm flood resilience is a restorative and adaptive learning organization in physical, natural, social, institutional and economic aspects within a certain range [16]. Liao Guixian et al. believe that storm flood resilience can be interpreted as the ability of a city to withstand floods, including the resistance ability of traditional flood prevention projects, flood adaptation organization ability to prevent casualties and property losses, and management ability to maintain urban infrastructure and social and economic stability [17].

### 2.2. Cloud Model

Cloud model is a cognitive model based on normal distribution function and membership function [18]. It is based on probability statistics and fuzzy set theory to realize the two-way conversion between qualitative concepts and quantitative data. It can effectively represent the concepts of fuzziness, randomness and uncertainty, and has been widely used in the fields of weight calculation, time series mining and decision evaluation [19]. Let U be a quantitative domain expressed by precise numerical values and C be a qualitative concept on the domain U. If the quantitative value is x ∈ U, and x is a random realization of the qualitative concept C, then the membership degree of x to C [0,1] is a random number with a stable tendency. Where there is X, the distribution of X in the domain U is called the subordinate cloud, which is called cloud for short. The result of a random numerical conversion of the qualitative concept through a specific algorithm using the digital characteristics of the cloud is called a cloud drop. A cloud drop contains an element X in the field U and its membership μ(x) degree to concept C in this conversion, denoted as (x, ‘μ(x)).

Normality is the most important property of the cloud model. Expectation Ex, entropy En and hyperentropy He are used to describe the three digital features of cloud model. A cloud model can usually be written as C = (Ex, En, He). Among them, the point that can best represent the qualitative concept of Expected Ex in the domain space is the most typical sample point for the quantization of the concept. Entropy En represents the measurable granularity of a qualitative concept. In general, the larger the entropy is, the more macroscopic the concept is. Entropy also reflects the uncertainty of the qualitative concept. Hyperentropy He represents the entropy of entropy and is a measure of entropy’s uncertainty. It reflects the randomness of samples representing qualitative concept values and reveals the association between fuzziness and randomness. In particular, when He = 0, the cloud model degenerates into the form of normal membership function.

Reverse cloud generator is a transformation model from quantitative value to qualitative concept. Its principle is that a certain number of precise values can be reasonably converted into appropriate qualitative language values (Ex, En, He) from statistical “cloud drops”. It is a mapping from quantitative to qualitative. This is a reverse and indirect process, which provides a favorable means for fuzzy comprehensive evaluation. Figure 1 is its schematic diagram.

### 2.3. TOPSIS

The approximate ideal solution ranking method is a commonly used and effective method in multi-objective decision analysis [20]. Its basic principle is to rank the evaluated objects based on the distance between positive and negative ideal solutions in multi-objective decision problems. If all the indicators of the positive ideal solution are optimal, it can be understood as a virtual optimal solution while the negative ideal solution is completely opposite. The TOPSIS method evaluates the relative merits and demerits of the evaluation object according to the distance memory ranking between the evaluation object and the idealized target. If the evaluation object is closest to the positive ideal solution, it is the optimal value; otherwise, it is the worst value [21]. The steps to calculate the degree of closeness by using the approximate ideal solution ordering method (TOPSIS) are as follows:

(1) Build the initial evaluation matrix

In this matrix, each row represents an index grade, and each column represents an evaluation index. Assume that there are n evaluation indexes and m index grades, then the initial evaluation matrix can be constructed as follows:
(1)$$A=\left[\begin{array}{ccccc} X_{11} & X_{12} & X_{31} & \cdots & X_{1n}\\ X_{21} & X_{22} & X_{32} & \cdots & X_{2n}\\ X_{31} & X_{32} & X_{33} & \cdots & X_{3n}\\ \cdots & \cdots & \cdots & & \\ X_{m1} & X_{m2} & X_{m3} & \cdots & X_{mn} \end{array}\right]$$


The element *x_ij_* in the initial evaluation matrix *A* can be expressed as the score value of the ith index relative to the j criteria.

Normalized numerical processing is carried out on the initial judgment matrix *A*, and then we can find the normalized initial judgment matrix of *A_ij_*:
(2)$$A_{ij}=x_{ij}/\sqrt{\sum_{j=1}^{m}x^{2}{}_{ij}},j=1,2,3\ldots m$$


(2) Construction of weighted standardized decision matrix

Using the cloud model and Delphi method, the relative weight *W* of each index factor is obtained.

Each column of the matrix *A_ij_* is multiplied by the weight *W_j_* to find the weighted normalization decision matrix *B_ij_*.
(3)$$B_{ij}=w_{j}A_{ij}$$


(3) Closeness degree analysis

The positive and negative ideal solutions *S*^+^ and *S*^−^ are calculated. They are respectively expressed as:
(4)$$S^{+}=\{\sum_{i}^{\max}B_{ij}/j\in J\},\{\sum_{i}^{\min}B_{ij}/j\in J^{\prime }\},i=1,2,3\ldots m$$


Here, $S^{+}$ represents the positive ideal solution, $B_{ij}$ represents the weighted standardized decision matrix, *J* = (*J* = 1, 2, 3 … *m*)/*j* represents benefit attribute, *J*′ = (*j* = 1, 2, 3 … *m*)/*j* represents the cost attribute.
(5)$$S^{-}=\{\sum_{i}^{\min}B_{ij}/j\in J\},\{\sum_{i}^{\max}B_{ij}/j\in J^{\prime }\},i=1,2,3\ldots m$$


Here, $S^{-}$ represents the negative ideal solution, $B_{ij}$ represents the weighted standardized decision matrix, *J* = (*J* = 1, 2, 3 … *m*)/*j* represents benefit attribute, *J*′ = (*j* = 1, 2, 3 … *m*)/*j* represents the cost attribute.

Calculate the distance between each evaluation index and the positive ideal solution *S_i_*^+^ and the negative ideal solution *S_i_*^−^. The calculation formula is as follows:
(6)$$S_{i}^{+}=\sqrt{\sum_{j=1}^{m}{\left(B_{ij}-S_{j}^{+}\right)}^{2}},i=1,2,\ldots ,m$$


Here, $S_{i}^{+}$ represents the distance between each evaluation index and the positive ideal solution, $B_{ij}$ represents the weighted standardized decision matrix, and $S_{j}^{+}$ represents the maximum value.
(7)$$S_{i}^{-}=\sqrt{\sum_{j=1}^{m}{\left(B_{ij}-S_{j}^{-}\right)}^{2}},i=1,2,\ldots ,m$$


Here, $S_{i}^{-}$ represents the distance between each evaluation index and the positive ideal solution, $B_{ij}$ represents the weighted standardized decision matrix, and $S_{j}^{-}$ represents the maximum value.

The closeness degree between each evaluation index grade and the positive ideal solution is calculated by the following formula:
(8)$$E_{i}=S_{i}^{-}/(S_{i}^{+}+S_{i}^{-})$$


## 3. Construction of Urban Stormwater Resilience Evaluation Index System

Based on relevant domestic and foreign literature [22,23,24,25,26,27] and the actual situation of major cities in China in dealing with rain flood damage, the evaluation index system of urban storm flood resilience was established from three dimensions of resistance, recovery and adaptability.

Resistance refers to the ability of a city to withstand the impact of rain and flood damage. Considering urban flood control facilities, for example, traditional urban flood control facilities only include some basic engineering facilities such as levees and dams, which do not meet the standards of building resilient cities. The construction of resilient cities not only requires more high-standard and strict urban flood control engineering facilities [23], but also requires urban residents to have self-consciousness, good flood resistance awareness, and certain personal flood control capacity [24]. Only by meeting the above conditions can we ensure that residents can react quickly and take corresponding flood control measures actively when the rain and flood damage comes, so as to reduce unnecessary economic losses and casualties.

Recovery refers to the ability of a city to change from an unbalanced state to a balanced state in time. It is intuitively shown as the ability of a city to eliminate the adverse effects caused by rain and flood (such as sewage, stagnant water, garbage and disease.) and restore the normal order of the city in a timely manner. Among them, urban infrastructure [23], residents’ education level [24], social capital [25], resource access [25], urban governance [26] and other factors will affect the speed and degree of urban recovery, which is an important factor to measure the resilience of a city.

Adaptability refers to a city’s ability to adjust itself to different levels of storm flooding. The city is formed by the interaction between man and nature, mutual feedback, through the establishment of good natural ecological system, and a developed city drainage system, and a new type of non-spot resource recycling system can improve the adaptability of a city for rain flood hazard, and among them, the good natural ecological system can effectively promote the regional climate and automatically adjust ability and the ability of ecological water culvert [27]. A developed urban drainage system has a certain degree of water storage capacity and can discharge urban rainwater to rivers, lakes and seas outside the city in a timely fashion. The new stormwater resource recycling system can make full use of stormwater resources through water storage and water purification, thus promoting the harmonious development of man and nature in the city.

Based on the above analysis, a set of evaluation index systems of urban resilience under the background of rain flood damage is constructed. The first-level index (target layer) is urban storm flood resilience, and the second-level index (decision-making layer) is resistance, resilience and adaptability. The third-level indicators are selected with reference to China Statistical Yearbook [28], with a total of 18 items, as shown in Table 1.

The selection basis of three-level indicators under resistance is that the level of education will affect the people’s awareness of flood fighting and flood control, broad and high-quality urban roads are conducive to rescue and evacuation in and after accidents, the action ability of the elderly and children is low, which will affect the emergency evacuation and rescue ability of the city, and flood control engineering facilities are important infrastructures to resist flood disasters

The selection basis of three-level indicators under recovery is that the employment rate will directly affect the city’s per capita GDP, and the city needs strong human and economic support to recover from the impact of rain and flood disasters. The level and ability of urban medical treatment will affect the rescue rate of the injured after rain and flood disasters.

The selection basis of three-level indicators under adaptability is that the good education level of urban residents will greatly promote the improvement of urban learning and adaptability. The urban green area and the water area of lakes and rivers will affect the ecological water cycle function and climate quality of the city, and the total urban water storage and comprehensive production capacity are conducive to the purification and treatment of rainwater.

## 4. Establishment of Urban Stormwater Resilience Evaluation Model

### 4.1. Cloud Model Combines with Delphi Method to Calculate the Weight of Indicators

The traditional index weight determination method is highly subjective. To reduce the subjectivity of weight determination, the cloud weight (Ex, En, He) method is selected in this paper to soften the traditional weight. Ex is expected to be the traditional weight, and entropy En and hyperentropy He are the parameters used to soften the weight.

(1) Delphi method

Relevant professionals and mostly experts and scholars were invited to weigh the indicators in the evaluation index system of urban rainwater toughness, and the importance of the lower index (such as the index layer) to the upper index (such as the criterion layer) was given a score based on 0–1.

(2) Cloud weight solution and correction

A reverse cloud generator algorithm is used to process the expert rating results to obtain the cloud weight corresponding to each index and generate the corresponding cloud map. The thicker the cloud in the cloud map, the higher the superentropy value and the greater the dispersion of cloud droplets, which indicates that the experts’ scoring results differ greatly, and the persuasion is low. Suggestions should be put forward on the experts’ scoring results and a new round of grading revision should be carried out to modify the index cloud weight. According to the procedure of “expert rating—result collection—cloud image judgment—opinion feedback—expert rating”, the cloud image is constantly improved to reduce the dispersion of cloud droplets, reduce the subjectivity of index cloud weight, and improve the accuracy and scientificity of evaluation results.

(3) Weight normalization processing

After calculating the index cloud weights of each layer of the urban stormwater toughness evaluation index system, it is necessary to normalize the index cloud weights of the same layer according to Equation (9).
(9)$$W_{ij}=\frac{w_{ij}}{\sum_{j=1}^{n}w_{ij}}$$
$w_{ij}$ is the cloud weight of indicators at each layer, $\sum_{j=1}^{n}w_{ij}$ is the sum of the cloud weight of indicators at the same layer, and $w_{ij}$ is the indicator weight after normalization.

The steps to determine the weight of each index by combining the cloud model and the Delphi method are as follows:

According to the steps to determine the weights of urban rainwater resilience assessment indicators in the previous section, 10 experts in the industry were invited to score the importance of the lower level indicators relative to the upper level, that is, the importance of B_i_(i = 1, 2, 3) relative to A, and C_i_ relative to B_I_. Ten points of the expert weight scoring table were issued and, ten copies were recycled. Finally, the cloud model is used to reverse the cloud generator to obtain the cloud weight of the lower layer relative to the upper index.

The evaluation index system of urban rainwater resilience involves a total of 18 indexes, which will not be described in this paper due to space limitation. Here, the determination process of the cloud weight between the resistance index of the criterion layer and the target layer is taken as an example to illustrate. The expert grading table is shown in Table 2.

Python programming language is used to implement reverse cloud generator algorithm, and the weight scoring results of B1 are processed to obtain the first score result’s cloud weight (0.9180, 0.0421, 0.0170). At the same time, the forward generator is used to generate the cloud map of the weight of B_1_ cloud, as shown in Figure 2. According to the weighted cloud map of B_1_ cloud, it can be seen that the cloud is thicker and the cloud drops are more dispersed, which means that there are big differences among experts when evaluating the importance of resistance relative to the toughness of urban rainwater in the scoring process. To improve the consistency of experts’ recognition of the cloud weight scoring results and reduce the subjective randomness of the scoring results, experts were invited to score again. After repeated feedback and correction, the cloud drop layer thickness in Figure 2 is obviously getting smaller and smaller, and the expert opinions are gradually unified. The cloud weight of resistance relative to urban stormwater resilience is (0.9230, 0.0163, 0.0065).

The above process is repeated to determine the index cloud weights of each layer of the urban storm flood resilience assessment index system and continuously revise them. Finally, the normalization process is conducted to obtain the index cloud weights of each layer of the urban storm flood resilience assessment index system, as shown in Table 3.

The index weights of each layer in the urban stormwater resilience assessment index system are summarized as shown in Table 4.

### 4.2. The Indicator Closeness Degree Was Calculated by TOPSIS Method

#### 4.2.1. Fuzzy Description of Resilience Indicator Grade

Because the units and orders of magnitude of each indicator are inconsistent, we set four ranges for each three-level indicator by referring to the standard ranges of each indicator in various provinces in China and assign corresponding scores to each range. Among them, 1 indicates that the index is lower than the national average, 2 and 3 indicate that the index is around the national average, and 4 indicates that the index is higher than the national average. We divided the fuzzy grade description of each resilience indicator into four levels, among which the first level indicates that the city has the lowest level of storm flood resilience, and the city under this level will face great flood risk and the subsequent recovery process will be very slow. Level 2 indicates that the level of storm flood resilience of a city is relatively poor. Cities under this level are still facing great flood risk, and uncontrollable accident consequences may occur when the rainstorm comes. Level 3 indicates that the city’s flood resilience level is in a good stage. Any city under this level has a certain resistance capacity and can be controlled in a relatively quick time once it is attacked by the flood. Level 4 indicates that the city has the highest degree of flood resilience, and a city under this level can make corresponding prediction measures in advance by virtue of its strong resistance and adaptability to avoid large losses.

Due to the limited space, only the fuzzy description of each third-level indicator under the second-level indicator is displayed. For the fuzzy description of the remaining third-level indicators. See the Appendix A.

(1) Fuzzy description of resistance indicator grade

The proportion of population over 60 years old and under 18 years old C_1_ value standard: the proportion of population over 60 years old and under 18 years old in the total population of the city. The fuzzy description of C_1_ is shown in Table 5.

(2) Fuzzy description of recovery indicator grade

Per capita GDP C_7_ value standard: measure the GDP ranking obtained by the National Bureau of Statistics to judge the level of the city’s per capita GDP. The fuzzy description of C_7_ is shown in Table 6.

(3) Fuzzy description of adaptability indicator grade

Per capita public green area C_12_ value standard: the average area of public green space occupied by each resident in the city. The fuzzy description of C_12_ is shown in Table 7.

(4) Secondary indicator evaluation grade value

The evaluation index grades of resistance are shown in Table 8.

The evaluation index grades of recovery are shown in Table 9.

The evaluation index grades of adaptability are shown in Table 10.

#### 4.2.2. Standard for Resilience Grade of Urban Stormwater

According to the grade fuzzy description of resilience index, the initial evaluation matrix A_K_ was constructed.

Set the resistance evaluation matrix as *A*_1_,
$$A_{1}=\left[\begin{array}{cccccc} 1 & 1 & 1 & 1 & 1 & 1\\ 2 & 2 & 2 & 2 & 2 & 2\\ 3 & 3 & 3 & 3 & 3 & 3\\ 4 & 4 & 4 & 4 & 4 & 4 \end{array}\right]$$


The distance S_1_^+^, S_1_^−^, and closeness degree E_i_^+^ between the resistance indicator class value and the positive and negative ideal solution were calculated, as Table 11 shown.

Set the recovery evaluation matrix as *A*_2_,
$$A_{2}=\left[\begin{array}{ccccc} 1 & 1 & 1 & 1 & 1\\ 2 & 2 & 2 & 2 & 2\\ 3 & 3 & 3 & 3 & 3\\ 4 & 4 & 4 & 4 & 4 \end{array}\right]$$


The distance S_1_^+^, S_1_^−^, and closeness degree E_i_^+^ between the recovery indicator class value and the positive and negative ideal solution were calculated, as Table 12 shown.

Set the adaptability evaluation matrix as *A*_3_,
$$A_{3}=\left[\begin{array}{ccccccc} 1 & 1 & 1 & 1 & 1 & 1 & 1\\ 2 & 2 & 2 & 2 & 2 & 2 & 2\\ 3 & 3 & 3 & 3 & 3 & 3 & 3\\ 4 & 4 & 4 & 4 & 4 & 4 & 4 \end{array}\right]$$


The distance S_1_^+^, S_1_^−^, and closeness degree E_i_^+^ between the adaptability indicator class value and the positive and negative ideal solution were calculated, as Table 13 shown.

To sum up, the urban stormwater resilience grade standard is shown in Table 14.

According to Table 14, the grading standards of urban stormwater resilience can be obtained as follows:

Level 1: 0 ≤ E_i_^+^ < 0.3896

Level 2: 0.3896 ≤ E_i_^+^ < 0.7194

Level 3: 0.7194 ≤ E_i_^+^ < 1

Level 4: E_i_^+^ = 1

In the first level, the resilience of urban stormwater is at the least ideal state. Once in the first level, the system will be faced with huge flood risk or the disaster resistance and recovery ability of the system is very poor, which will cause extremely serious consequences. Level 2 refers to the situation in which the resilience of urban stormwater is relatively poor. The system in level 2 is still facing great flood risk and will cause uncontrollable accident consequences once the rainstorm occurs. Level 3 refers to the situation in which the resilience of urban stormwater is relatively good. The system in level 3 is capable of resisting floods to a certain extent, and it can be controlled within a certain period of time once the rainstorm occurs. Level 4 is the ideal condition of urban stormwater resilience. The system in level 4 can predict and avoid or respond to the flood in time, so that it will not cause great losses. The system has a strong learning ability and can draw experience from the flood disaster suffered by the system, which makes the resilience level of the system higher and higher.

## 5. Example Analysis and Discussion

Taking the heavy rainstorm in Beijing on 7.21 as an example, this paper comprehensively evaluated the rainwater toughness of Beijing at that time by applying the cloud model TOPSIS resilience evaluation model, found out the weak links in the response to the rainwater and flood damage of Beijing at that time and put forward corresponding suggestions.

### 5.1. Rainstorm Flood Resilience Indicator Grade of Beijing

Through referring to relevant literature, we have mastered the relevant data of Beijing before the heavy rainstorm on July 21, and now we have classified the three-level indicators of rainstorm resilience in Beijing, as Table 15 shown.

### 5.2. Stormwater Resilience Assessment in Beijing

According to Formula (1), the initial evaluation matrix *A* of stormwater resilience in Beijing is obtained.
$$A=\left[\begin{array}{cccccccccccccccccc} 1 & 1 & 1 & 1 & 1 & 1 & 1 & 1 & 1 & 1 & 1 & 1 & 1 & 1 & 1 & 1 & 1 & 1\\ 2 & 2 & 2 & 2 & 2 & 2 & 2 & 2 & 2 & 2 & 2 & 2 & 2 & 2 & 2 & 2 & 2 & 2\\ 3 & 3 & 3 & 3 & 3 & 3 & 3 & 3 & 3 & 3 & 3 & 3 & 3 & 3 & 3 & 3 & 3 & 3\\ 4 & 4 & 4 & 4 & 4 & 4 & 4 & 4 & 4 & 4 & 4 & 4 & 4 & 4 & 4 & 4 & 4 & 4\\ 2 & 4 & 3 & 1 & 2 & 2 & 3 & 4 & 3 & 4 & 2 & 4 & 4 & 1 & 3 & 1 & 1 & 3 \end{array}\right]$$


Then, the initial evaluation matrix *A* is dimensionless to obtain the standard decision matrix *B*.
$$B=\left[\begin{array}{ccccccccccc} 0 & 0 & 0 & 0 & 0 & \cdots & 0 & 0 & 0 & 0 & 0\\ 1/3 & 1/3 & 1/3 & 1/3 & 1/3 & \cdots & 1/3 & 1/3 & 1/3 & 1/3 & 1/3\\ 2/3 & 2/3 & 2/3 & 2/3 & 2/3 & \cdots & 2/3 & 2/3 & 2/3 & 2/3 & 2/3\\ 1 & 1 & 1 & 1 & 1 & \cdots & 1 & 1 & 1 & 1 & 1\\ 1/3 & 1 & 2/3 & 0 & 1/3 & \cdots & 0 & 2/3 & 0 & 0 & 2/3 \end{array}\right]$$


According to Formula (4) to Formula (8), the distance S_1_^+^, S_1_^−^, and the closeness degree E_1_^+^ between the value of stormwater resilience capacity indicator grade in Beijing and the positive and negative ideal solutions are obtained, as shown in Table 16.

Through calculation, the final result of the stormwater resilience assessment of Beijing is 0.5196, which means that the stormwater resilience level of Beijing is at the second level. In other words, the stormwater resilience level of Beijing is relatively poor. Once a flood with large precipitation occurs, uncontrollable accident consequences will be caused.

It can be seen from the fuzzy description of the stormwater resilience indicator of Beijing above that the indicator score of the pumping and discharging capacity of pumps, the surface area of lakes and rivers, the capacity of reservoirs and the comprehensive production capacity of water plants in the central urban area of Beijing is only 1. This indicates that the water storage of Beijing is low, and the capacity of self-production of water and the drainage capacity of the urban system are at the downstream level. In recent years, although projects such as “South-to-North Water Diversion” have been carried out to supply water from water-rich areas to water-poor cities, when extreme weather disasters such as 7.21 come, cities with low stormwater resilience still find it difficult to resist the impact and pressure brought about by disasters, and bear relatively serious consequences. This reflects that the key for a city to cope with the hazards of natural disasters is to improve the resilience of the city itself.

According to the above numerical results and analysis, it is suggested that the relevant urban management departments should improve the pumping and discharging capacity of urban water pumps and the comprehensive production capacity of water plants, so as to strengthen the drainage capacity of urban systems and increase the total water storage of the city and at the same time, strengthen the protection of urban rivers and waters, publicize the importance of water resources protection and encourage urban people to actively participate in the protection of urban water resources, so as to avoid the drying up and degradation of urban rivers and waters to the greatest extent. It is believed that with the effective implementation of these measures, the resilience level of urban stormwater will be significantly improved.

## 6. Conclusions

(1) According to the safety assessment methods developed by our predecessors and relevant literature and data, according to the actual situation of urban flood resistance and relevant national regulations, the indicators affecting urban rain and flood resilience are analyzed, and a three-level indicator system is constructed. The first-level indicators are urban rain and flood resilience, the second-level indicators are resistance, recovery and adaptability, and there are 18 third-level indicators in total. The indicator system of the model is relatively perfect, and the weight of each evaluation indicator is effectively unified.

(2) The cloud model and Delphi method are used to obtain the relative weight of evaluation indicators at all levels. The relative weight value indicates the influence degree of each subordinate indicator on the final goal. By comparing the relative weight value, we can find the vulnerable points in the overall system, and then provide targeted management measures and suggestions for relevant departments.

(3) Through the fuzzy description of the resilience indicator level and using the approximate ideal solution ranking method (TOPSIS method), the distance and closeness between each level indicator and the positive and negative ideal solution can be obtained. The calculated closeness can be used to represent the high or low degree of urban stormwater resilience.

(4) The constructed cloud TOPSIS evaluation model is applied to Beijing under the Severe Rainstorm on July 21. Through an all-round data search, sorting and calculation, it is finally determined that the stormwater resilience of Beijing was at the second level at that time.

Due to a relatively small number of quantitative indicators, which fail to cover all aspects that affect urban resilience, and individual indicators are limited by regions, the applicability of the model established in this paper is limited. The next step is to construct an indicator system from multi-dimensional considerations to improve the applicability of the model. In addition, the characteristics of this paper that are absolute values are used to indicate that it can contrast the evaluation indicator and the national average gap, but considering that the relative characteristics are used to indicate if some indicators can be according to different population scales, area size, and organizational structure, the results of the city are also needed to be considered in future research work.

## Figures and Tables

**Figure 1 ijerph-19-00038-f001:**
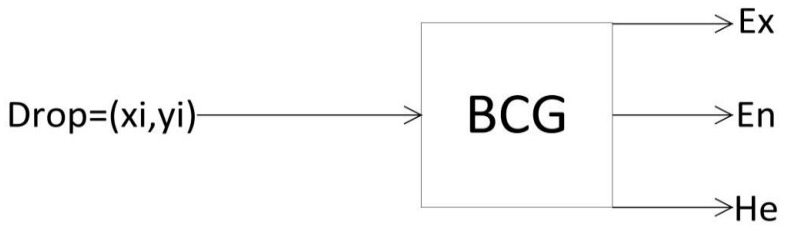
Schematic diagram of reverse cloud generator.

**Figure 2 ijerph-19-00038-f002:**
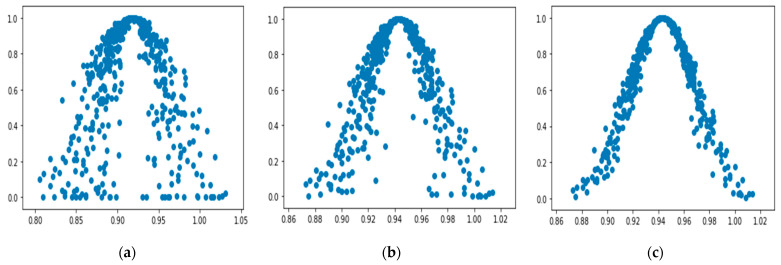
Process diagram of cloud weight correction. (**a**) Experts score for the first time; (**b**) experts score for the second time; (**c**) experts score for the third time.

**Table 1 ijerph-19-00038-t001:** Urban stormwater resilience evaluation index system.

First-Level Indicators	Secondary Indicators	Tertiary Indicators
Urban stormwater resilience	Resistance	Percentage of the population aged over 60 and under 18
		Length of levee
		Length of drainage pipe
		Central city pump pumping capacity
		Per capita road area
		The average number of students in institutions of higher learning per 10,000 population
	Recovery	GDP per capita
		The employment rate
		Urban health insurance coverage
		Number of sanitary beds per 1000 population
		Number of health workers per 1000 population
	Adaptability	Per capita public green area
		Green coverage rate in built-up areas
		The surface area of lakes and rivers
		Paddy farmland area
		The reservoir capacity
		Comprehensive production capacity of water plant
		Sewage treatment capacity

**Table 2 ijerph-19-00038-t002:** Weight grading table of B_i_ relative to A.

Expert	1	2	3	4	5	6	7	8	9	10
weight	0.9	0.98	0.96	0.96	0.9	0.88	0.9	0.87	0.94	0.89

**Table 3 ijerph-19-00038-t003:** Urban stormwater resilience assessment index system cloud weight.

The Target Layer	Rule Layer		Index Layer	
	Indicators	Cloud Weight	Indicators	Cloud Weight
A	B_1_	(0.452, 0.0113, 0.0032)	C_1_	(0.068, 0.0215, 0.0058)
			C_2_	(0.154, 0.0265, 0.0049)
			C_3_	(0.231, 0.0289, 0.0037)
			C_4_	(0.167, 0.0254, 0.0042)
			C_5_	(0.136, 0.0312, 0.0029)
			C_6_	(0.244, 0.0268, 0.0036)
	B_2_	(0.216, 0.0014, 0.0036)	C_7_	(0.236, 0.0196, 0.0080)
			C_8_	(0.186, 0.0211, 0.0041)
			C_9_	(0.323, 0.0266, 0.0045)
			C_10_	(0.145, 0.0298, 0.0071)
			C_11_	(0.110, 0.0184, 0.0064)
	B_3_	(0.332, 0.0165, 0.0027)	C_12_	(0.163, 0.0259, 0.0043)
			C_13_	(0.222, 0.0241, 0.0038)
			C_14_	(0.223, 0.0278, 0.0056)
			C_15_	(0.132, 0.0321, 0.0071)
			C_16_	(0.076, 0.0224, 0.0052)
			C_17_	(0.086, 0.0235, 0.0068)
			C_18_	(0.098, 0.0251, 0.0025)

**Table 4 ijerph-19-00038-t004:** Weight table of urban stormwater resilience evaluation indicators.

	B Layer and Weight	Resistance	Recovery	Adaptability	Total Ranking Weight of C Layer Factors
C Layer		0.452	0.216	0.332	
Percentage of the population aged over 60 and under 18 C_1_		0.068	0.000	0.000	0.0307
Length of levee C_2_		0.154	0.000	0.000	0.0696
Length of drainage pipe C_3_		0.231	0.000	0.000	0.1044
Central city pump pumping capacity C_4_		0.167	0.000	0.000	0.0755
Per capita road area C_5_		0.136	0.000	0.000	0.0615
The average number of students in institutions of higher learning per 10,000 population C_6_		0.244	0.000	0.000	0.1103
GDP per capita C_7_		0.000	0.236	0.000	0.0509
The employment rate C_8_		0.000	0.186	0.000	0.0402
Urban health insurance coverage C_9_		0.000	0.323	0.000	0.0698
Number of sanitary beds per 1000 population C_10_		0.000	0.145	0.000	0.0313
Number of health workers per 1000 population C_11_		0.000	0.110	0.000	0.0238
Per capita public green area C_12_		0.000	0.000	0.163	0.0541
Green coverage rate in built-up areas C_13_		0.000	0.000	0.222	0.0737
The surface area of lakes and rivers C_14_		0.000	0.000	0.223	0.0740
Paddy farmland area C_15_		0.000	0.000	0.132	0.0438
The reservoir capacity C_16_		0.000	0.000	0.076	0.0252
Comprehensive production capacity of water plant C_17_		0.000	0.000	0.086	0.0286
Sewage treatment capacity C_18_		0.000	0.000	0.098	0.0326

**Table 5 ijerph-19-00038-t005:** Fuzzy description of the percentage of the population aged over 60 and under 18.

Percentage of the Population Aged Over 60 and Under 18 C_1_	Scoring Criteria	Score
Level 1: The worst plan Y^−^	Percentage > 30%	1
Level 2	Percentage between 20% and 30%	2
Level 3	Percentage between 10% and 20%	3
Level 4: The best plan Y^+^	Percentage < 10%	4

**Table 6 ijerph-19-00038-t006:** Fuzzy description of GDP per capita.

GDP Per Capita C_7_	Scoring Criteria	Score
Level 1: The worst plan Y^−^	GDP per capita < JPY 50,000	1
Level 2	GDP per capita between JPY 50,000 and 75,000	2
Level 3	GDP per capita between JPY 75,000 and 100,000	3
Level 4: The best plan Y^+^	GDP per capita > JPY 100,000	4

**Table 7 ijerph-19-00038-t007:** Fuzzy description of per capita public green area.

Per Capita Public Green Area C_12_	Scoring Criteria	Score
Level 1: the worst plan Y^−^	Per capita public green area < 6 m^2^	1
Level 2	Per capita public green area between 6 m^2^ and 7 m^2^	2
Level 3	Per capita public green area between 7 m^2^ and 8 m^2^	3
Level 4: the best plan Y^+^	Per capita public green area > 8 m^2^	4

**Table 8 ijerph-19-00038-t008:** Values of resistance evaluation indicator grades.

Indicator Level	Level 1	Level 2	Level 3	Level 4
Percentage of the population aged over 60 and under 18 C_1_	1	2	3	4
Length of levee C_2_	1	2	3	4
Length of drainage pipe C_3_	1	2	3	4
Central city pump pumping capacity C_4_	1	2	3	4
Per capita road area C_5_	1	2	3	4
The average number of students in institutions of higher learning per 10,000 population C_6_	1	2	3	4

**Table 9 ijerph-19-00038-t009:** Values of recovery evaluation indicator grades.

Indicator Level	Level 1	Level 2	Level 3	Level 4
GDP per capita C_7_	1	2	3	4
The employment rate C_8_	1	2	3	4
Urban health insurance coverage C_9_	1	2	3	4
Number of sanitary beds per 1000 population C_10_	1	2	3	4
Number of health workers per 1000 population C_11_	1	2	3	4

**Table 10 ijerph-19-00038-t010:** Values of adaptability evaluation indicator grades.

Indicator Level	Level 1	Level 2	Level 3	Level 4
Per capita public green area C_12_	1	2	3	4
Green coverage rate in built-up areas C_13_	1	2	3	4
The surface area of lakes and rivers C_14_	1	2	3	4
Paddy farmland area C_15_	1	2	3	4
The reservoir capacity C_16_	1	2	3	4
Comprehensive production capacity of water plant C_17_	1	2	3	4
Sewage treatment capacity C_18_	1	2	3	4

**Table 11 ijerph-19-00038-t011:** Resistance grade standard.

Resistance Level	S_1_^+^	S_1_^−^	E_i_^+^
Level 1	0.4332	0	0
Level 2	0.2888	0.1444	0.3332
Level 3	0.1444	0.2888	0.6537
Level 4	0	0.4332	1

**Table 12 ijerph-19-00038-t012:** Recovery grade standard.

Recovery Level	S_2_^+^	S_2_^−^	E_i_^+^
Level 1	0.4772	0	0
Level 2	0.3182	0.1591	0.3061
Level 3	0.1591	0.3182	0.7256
Level 4	0	0	1

**Table 13 ijerph-19-00038-t013:** Adaptability grade standard.

Adaptability Level	S_3_^+^	S_3_^−^	E_i_^+^
Level 1	0.4072	0	0
Level 2	0.2714	0.1357	0.3692
Level 3	0.1357	0.2714	0.7143
Level 4	0	0.4072	1

**Table 14 ijerph-19-00038-t014:** Urban stormwater resilience grade standard.

Resilience Level	S^+^	S^−^	E_i_^+^
Level 1	0.2814	0	0
Level 2	0.1876	0.0938	0.3896
Level 3	0.0938	0.1867	0.7194
Level 4	0	0.2814	1

**Table 15 ijerph-19-00038-t015:** Grade classification of stormwater resilience indicators in Beijing.

Indicator	Actual Description	Score
Percentage of the population aged over 60 and under 18 C_1_	In Beijing, the population aged over 60 and under 18 accounts for about 24% of the total registered population	2
Length of levee C_2_	Total length of levee > 20 km	4
Length of drainage pipe C_3_	The total length of the drainage pipe is about 10,200 km	3
Central city pump pumping capacity C_4_	Most pumps in the central urban area have a recurrence period of less than 2 years, which is difficult to cope with extreme weather standards	1
Per capita road area C_5_	The road area per capita is about 7.53 m^2^	2
The average number of students in institutions of higher learning per 10,000 population C_6_	The average number of students in institutions of higher learning is close to 1, 100 per 10,000 population	2
GDP per capita C_7_	Beijing’s per capita GDP in 2012 was JPY 86,024.26	3
Employment rate C_8_	The employment rate in Beijing was 96.1% in 2012	4
Urban health insurance coverage C_9_	The coverage of urban medical insurance is close to 85%	3
Number of sanitary beds per 1000 population C_10_	The number of sanitary beds is about 9.5 per 1000 population	4
Number of health workers per 1000 population C_11_	The number of health workers is about 11.39 per 1000 population	2
Per capita public green area C_12_	The per capita public green area is about 10.23 m^2^	4
Green coverage rate in built-up areas C_13_	The green coverage rate of the built-up area is about 46.25%	4
The surface area of lakes and rivers C_14_	The surface area of lakes and rivers is about 6.88 km^2^	1
Paddy farmland area C_15_	The cultivated area of paddy field is about 2200 km^2^	3
The reservoir capacity C_16_	At the end of the year, the total amount of water stored in 18 large and medium-sized reservoirs in the city was about 15.06 billion m^3^	1
Comprehensive production capacity of water plant C_17_	The combined production capacity of the water plant is about 4,900,000 t	1
Sewage treatment capacity C_18_	Sewage treatment plants treat about 3,000,000 m^3^ of sewage every day	3

**Table 16 ijerph-19-00038-t016:** Urban stormwater resilience grade standard.

Resilience Level	S^+^	S^−^	E_i_^+^
Level 1	0.2814	0	0
Level 2	0.1876	0.0938	0.3896
Level 3	0.0938	0.1867	0.7194
Level 4	0	0.2814	1
BeiJing	0.1612	0.1743	0.5196

## Data Availability

Not applicable.

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
