# Peer review of "Urban Stormwater Resilience Assessment Method Based on Cloud Model and TOPSIS"

_ijerph, 2021, doi:10.3390/ijerph19010038_

Round 1

Reviewer 1 Report

In this paper, we use the Cloud-TOPSIS model to evaluate the resilience of cities to heavy rainfall. This method is useful to evaluate the resilience of various cities. No significant changes are needed for this paper.

Minor corrections are listed below

  1. (7) equation

is a mistake for , isn’t it?

  1. Line 304

Isn't Figure.3 a mistake for Figure.2?

  1. Figure 2.

The resolution of the Figure is low, please increase the resolution of the image.

  1. About Fuzzy Description

In the case of quantitative evaluation, such as the length of levees and sewerage systems, the evaluation standard is likely to differ depending on the size of the city. Therefore, in the case of quantitative evaluation, we should not use absolute values, but rather standardized values, so that the method can be more generalized. Since this is very important when considering the applicability of this method, please explain it.

Author Response

Response to Reviewer 1 Comments

Point 1: (7) equation is a mistake for , isnt it?

Response 1:Yes, there is a slight problem with (7) equation, which has been modified.

Point 2:Line 304.

Isn't Figure.3 a mistake for Figure.2?

Response 2:Figure 3 has been changed to Figure 2 in this article.

Point 3:Figure 2.The resolution of the Figure is low, please increase the resolution of the image.

Response 3:The resolution of the image has been increased.

Point 4:About Fuzzy Description

In the case of quantitative evaluation, such as the length of levees and sewerage systems, the evaluation standard is likely to differ depending on the size of the city. Therefore, in the case of quantitative evaluation, we should not use absolute values, but rather standardized values, so that the method can be more generalized. Since this is very important when considering the applicability of this method, please explain it.

Response 4:All fuzzy description values in this paper are taken as the standard of each grade after the data survey of each province, and this value can measure the stage of an index in a city.

Reviewer 2 Report

It is necessary to revise.

  • Please do not use “in order to” in the manuscript.
  • Line 90, please add a dot after “al”.
  • Please delete all “so on, etc." from the manuscript.
  • In Figure 1, please define BCG and Drop.
  • Line 150, please delete comma before “while.”
  • Line 168, please explain the Delphi method.
  • Please, describe closeness degree analysis.
  • Please define all parameters of equations 4,5, and 6 under each equation with the unit of measurement.
  • Line 192, please change “take” to “considering.”
  • Line 193, why did you use capital letters for “DAMS”.
  • Line 197, please add a comma before” and.”
  • Line 198, why did you use” Can we”.
  • So many definitions of a resilient city exist between lines 187 to 224. Please shorten the text.
  • Line 232, please do not use “based on the fact that.”
  • Please define the parameter of Eq. 9 under the equation.
  • Line 280 to 283 are repetition sentences. Please delete it.
  • How do you prove 10 experts filling your request enough?
  • Line 295, any extra information should be put to supplementary file.
  • The number of Tables is high and it should be reduced and if necessary put to supplementary file.
  • Before the conclusion subtitle, it is necessary to compare your work with a similar one and indicate the novelty.
  • Line 539 to 556 are not discussion part, it is the limitation of your study. The discussion part should be after the result before the conclusions subtitle.
  • Line 518 to 529 are key results.
  • The conclusion should be rewritten and must be the key results of your study.
  • The authors studied Beijing city as a case study; however, there are not any key results of the resilience of the city.

Author Response

Response to Reviewer 2 Comments

Point 1:Please do not use "in order to" in the manuscript.

Response 1:All "in order to" has been replaced in the manuscript.

Point 2:Line 90, please add a dot after "al".

Response 2:It was dotted after "al".

Point 3:Please delete all "so on, etc." from the manuscript.

Response 3:All "so on, etc." has been deleted from the manuscript.

Point 4:In Figure 1, please define BCG and Drop.

Response 4:BCG stands for reverse transport generator and Drop stands for cloud Drop

Point 5:Line 150, please delete comma before "while".

Response 5:The comma has been removed before "while".

Point 6:Line 168, please explain the Delphi method.

Response 6:Delphi method on the basis of the system program, with the method of anonymous comments, which shall not discuss each other between experts, not occur lateral connection, can only be in a relationship with investigators, by experts on the rounds survey questionnaire questions, after consultation, inductive, modify repeatedly, finally together into experts are basically identical, as the result of forecast. This method is widely representative and reliable.

Point 7:Please, describe closeness degree analysis.

Response 7:Closeness degree represents the distance between each evaluation target and the best evaluation target and the worst evaluation target. The value of closeness degree is between 0 and 1. The closer the value is to 1, the closer the corresponding evaluation target is to the optimal level. On the contrary, the closer the value is to 0, the closer the evaluation target is to the worst level, which can be used as the basis to evaluate the quality of the target.

Point 8:Please define all parameters of equations 4,5, and 6 under each equation with the unit of measurement.

Response 8:The meanings of each parameter have been explained in equation 4, 5, 6 and 7.

Point 9:Line 192, please change "take" to "considering".

Response 9:I have changed "take" to "considering" in the manuscript

Point 10:Line 193, why did you use capital letters for "DAMS".

Response 10:dams have been represented in lower case.

Point 11:Line 197, please add a comma before "and".

Response 11:A comma has been added before "and".

Point 12:Line 198, why did you use "Can we".

Response 12:From the perspective of a manager, we consider various evaluation indicators of urban storm flood resilience

Point 13:So many definitions of a resilient city exist between lines 187 to 224. Please shorten the text.

Response 13:The definition of a resilient city has been shortened.

Point 14:Line 232, please do not use "based on the fact that."

Response 14:It has been revised in the manuscript.

Point 15:Please define the parameter of Eq. 9 under the equation.

Response 15:The meanings of each parameter have been explained in equation 9.

Point 16:Line 280 to 283 are repetition sentences. Please delete it.

Response 16:Repetitions have been excised from the manuscript.

Point 17:How do you prove 10 experts filling your request enough?

Response 17:In The Delphi method, it is reasonable to invite about 10 experts, and these experts get the result after scoring at least three times. This data is relatively reliable

Point 18:Line 295, any extra information should be put to supplementary file.

Response 18:I think this part can be put in the body of the manuscript, it is not clear what you mean by the extra information.

Point 19:The number of Tables is high and it should be reduced and if necessary put to supplementary file.

Response 19:The number of tables has been reduced.

Point 20:Before the conclusion subtitle, it is necessary to compare your work with a similar one and indicate the novelty.

Response 20:Because the time for this modification is limited, we will modify it according to your requirements later.

Point 21:Line 539 to 556 are not discussion part, it is the limitation of your study. The discussion part should be after the result before the conclusions subtitle.

Response 21:The discussion has been placed before the conclusion and after the analysis of the results.

Point 22:Line 518 to 529 are key results.

The conclusion should be rewritten and must be the key results of your study.

Response 22:The conclusion has been modified and improved to add the limitations of this study and the direction of future research.

Point 23:The authors studied Beijing city as a case study; however, there are not any key results of the resilience of the city.

Response 23:In this paper, the value of storm-flood toughness in Beijing is calculated in Section 5.2, and the toughness grade is at the second level.

Reviewer 3 Report

I have read with interest your paper on the Urban stormwater resilience assessment method based on cloud model and TOPSIS. The article is quite well organized, however, I have a few comments:

Abstract is written too vaguely from methodical point of view, it lacks the investigation idea, way of examination and conclusions as well as needs style correction

Most keywords duplicate the title of the paper, it should be avoided

Selection of tertiary indicators (Tab. 1) is obviously subjective, they are well described and generally acceptable. However:

  1. it should be described what was the reasons (why such numbers) for dividing particular indicators into score classes (Tab. 5-22),
  2. using the paddy farmland areas as indicator, limits seriously from regional point of view applicability of presented method. It should be clearly mentioned in method introduction and in conclusions,
  3. some of tertiary indicators are not expressed as relative characteristics (Tab. 6, 7, 8, 18, 19, 20, 21, 22). Phenomena and processes estimated by them depend significantly on size, structure and morphology of the city and local hydroclimatic conditions. For example, total length of the levee should be referred to river network (e.g. in % or km/km), length of the drainage pipe to city area size (e.g. in km/km2) etc. Using the absolute units instead of relative in these indicators makes results incomparable for cites with different size, morphology and demographical structure. If authors are going to keep these indicators in presented formulas, all assumptions and limitations must be described in detail in methodical part and summarized in conclusions.

Conclusions are too vaguely, similarly to abstract. If indicator system is ‘perfect’ and method works ‘effectively’ (line 524) will be known after comparable studies for different cities and referring results to some standards. As I suppose, authors idea was presentation of the method, not its application (Beijing example is an illustration of calculation only). As a result, much more attention should be paid to discussion about properties and consequences of subsequent stages of the procedure form methodical and practical point of view. Therefore, I advise separate chapter ‘Discussions’ with problems mentioned above and serious reconstruction of the ‘Conclusions’ chapter.

Author Response

Response to Reviewer 3 Comments

Point 1:Abstract is written too vaguely from methodical point of view, it lacks the investigation idea, way of examination and conclusions as well as needs style correction  

Response 1:The abstract section has been rewritten.

Point 2:Most keywords duplicate the title of the paper, it should be avoided.

Response 2:The keywords have been modified.

Point 3:it should be described what was the reasons (why such numbers) for dividing particular indicators into score classes (Tab. 5-22),

Response 3:The reasons for the classification of specific indicators have been described.

Point 4:using the paddy farmland areas as indicator, limits seriously from regional point of view applicability of presented method. It should be clearly mentioned in method introduction and in conclusions.

Response 4:The regional limitations of the indicators have been explained in the manuscript.

Point 5:some of tertiary indicators are not expressed as relative characteristics (Tab. 6, 7, 8, 18, 19, 20, 21, 22). Phenomena and processes estimated by them depend significantly on size, structure and morphology of the city and local hydroclimatic conditions. For example, total length of the levee should be referred to river network (e.g. in % or km/km), length of the drainage pipe to city area size (e.g. in km/km2) etc. Using the absolute units instead of relative in these indicators makes results incomparable for cites with different size, morphology and demographical structure. If authors are going to keep these indicators in presented formulas, all assumptions and limitations must be described in detail in methodical part and summarized in conclusions.

Response 5:The absolute characteristics selected in this paper are based on the average level of domestic cities, which has been explained in this part.

Point 6:Conclusions are too vaguely, similarly to abstract. If indicator system is ‘perfect’ and method works ‘effectively’ (line 524) will be known after comparable studies for different cities and referring results to some standards. As I suppose, authors idea was presentation of the method, not its application (Beijing example is an illustration of calculation only). As a result, much more attention should be paid to discussion about properties and consequences of subsequent stages of the procedure form methodical and practical point of view. Therefore, I advise separate chapter ‘Discussions’ with problems mentioned above and serious reconstruction of the ‘Conclusions’ chapter.

Response 6:The "discussion" and "conclusion" have been explained separately, and the conclusion part has been reconstructed.

Round 2

Reviewer 3 Report

I accept corrected version if the manuscript